# Epidemiological Patterns of Seasonal Respiratory Viruses during the COVID-19 Pandemic in Madagascar, March 2020–May 2022

**DOI:** 10.3390/v15010012

**Published:** 2022-12-20

**Authors:** Norosoa Harline Razanajatovo, Tsiry Hasina Randriambolamanantsoa, Joelinotahiana Hasina Rabarison, Laurence Randrianasolo, Miamina Fidy Ankasitrahana, Arvé Ratsimbazafy, Antso Hasina Raherinandrasana, Helisoa Razafimanjato, Vololoniaina Raharinosy, Soa Fy Andriamandimby, Jean-Michel Heraud, Philippe Dussart, Vincent Lacoste

**Affiliations:** 1National Influenza Center, Virology Unit, Institut Pasteur of Madagascar, Antananarivo 101, Madagascar; 2Epidemiology and Clinical Research Unit, Institut Pasteur of Madagascar, Antananarivo 101, Madagascar; 3Direction de la Veille Sanitaire, de la Surveillance Epidémiologique et Ripostes, Ministry of Public Health, Antananarivo 101, Madagascar; 4Virology Department, Institut Pasteur de Dakar, Dakar 12900, Senegal

**Keywords:** viruses, SARS-CoV-2, epidemics, epidemiology, etiology, detection, Madagascar, Africa

## Abstract

Three epidemic waves of coronavirus disease-19 (COVID-19) occurred in Madagascar from March 2020 to May 2022, with a positivity rate of severe acute respiratory syndrome coronavirus 2 (SARS-CoV-2) of 21% to 33%. Our study aimed to identify the impact of COVID-19 on the epidemiology of seasonal respiratory viruses (RVs) in Madagascar. We used two different specimen sources (SpS). First, 2987 nasopharyngeal (NP) specimens were randomly selected from symptomatic patients between March 2020 and May 2022 who tested negative for SARS-CoV-2 and were tested for 14 RVs by multiplex real-time PCR. Second, 6297 NP specimens were collected between March 2020 and May 2022 from patients visiting our sentinel sites of the influenza sentinel network. The samples were tested for influenza, respiratory syncytial virus (RSV), and SARS-CoV-2. From SpS-1, 19% (569/2987) of samples tested positive for at least one RV. Rhinovirus (6.3%, 187/2987) was the most frequently detected virus during the first two waves, whereas influenza predominated during the third. From SpS-2, influenza, SARS-CoV-2, and RSV accounted for 5.4%, 24.5%, and 39.4% of the detected viruses, respectively. During the study period, we observed three different RV circulation profiles. Certain viruses circulated sporadically, with increased activity in between waves of SARS-CoV-2. Other viruses continued to circulate regardless of the COVID-19 situation. Certain viruses were severely disrupted by the spread of SARS-CoV-2. Our findings underline the importance and necessity of maintaining an integrated disease surveillance system for the surveillance and monitoring of RVs of public health interest.

## 1. Introduction

Since its emergence in late 2019, severe acute respiratory syndrome coronavirus 2 (SARS-CoV-2) has spread throughout the world, infecting billions of people and causing millions of deaths [1]. Madagascar has not been spared by the coronavirus disease-19 (COVID-19) pandemic. The first detection of SARS-CoV-2 cases in Madagascar was reported on March 19, 2020 [2]. Between March and September 2020, the country experienced the first epidemic wave of COVID-19. To limit disease transmission, the authorities implemented mitigation strategies, including non-pharmacological intervention (NPI) measures, such as lockdowns, travel bans, social distancing, the permanent wearing of facial masks, handwashing, and school closures. During this period, the Institut Pasteur de Madagascar (IPM) tested 26,415 nasopharyngeal (NP) swabs from (i) suspected COVID-19 cases, (ii) travelers returning from affected areas, and (iii) the contacts of laboratory cases confirmed by real-time RT-PCR. Overall, approximately 21% turned out to be laboratory-confirmed SARS-CoV-2 cases [2]. A second wave occurred from March to May 2021, during which the rate of positivity for SARS-CoV-2 was approximately 29%. Finally, a third wave occurred between December 2021 and February 2022, yielding a detection rate of 33%. Recent studies have shown that the early epidemic periods of SARS-CoV-2 had a significant impact on the circulation pattern of seasonal respiratory viruses (RVs). In addition, the activity of certain viruses with a seasonal profile was delayed [3,4,5,6,7]. Given that the clinical signs and symptoms of COVID-19 are similar to those of RVs, laboratory testing is required for differential diagnosis. In addition, determining trends in RV circulation after the emergence of SARS-CoV-2 is essential for adapting the surveillance system at a national and international level. Our study aimed to (i) identify viral etiologies associated with acute respiratory infection (ARI) in the context of SARS-CoV-2 circulation in Madagascar (from March 2020 to May 2022) and (ii) describe the epidemiology of RVs during the pre-pandemic and pandemic periods.

## 2. Methods

### 2.1. Study Design

We used two different specimen sources (SpS) to conduct our study (Appendix A). First, we randomly selected 2987 NP swabs (SpS-1) from 20,254 SARS-CoV-2-negative specimens collected from symptomatic cases between March 2020 and February 2022, a period covering the first three epidemic waves of COVID-19 in Madagascar. These specimens originated from various regions of Madagascar. Methodologies for suspected COVID-19 case sampling and SARS-CoV-2 testing were recently published [2]. The randomly selected samples were tested using five multiplex real-time PCR (rtPCR) assays to detect 14 RVs, including influenza (FLU) A and B, respiratory syncytial virus (RSV), human metapneumovirus (HMPV), rhinovirus (HRV), human coronavirus (CoV)-OC43, 229E, NL63, HKU1, parainfluenza viruses (PIV)-1, -2, -3, adenovirus (ADV), and bocavirus (BOV). These five rtPCR assays have already been extensively described [8]. For the second source of specimens (SpS-2), we selected all the NP swabs collected from patients presenting with an influenza-like illness (ILI) or severe acute respiratory infection (SARI) at our healthcare centers or clinics of the Influenza Surveillance Network (ISN). The ISN has been in place for years and monitors the circulation of influenza and other RVs throughout the country through ILI and SARI surveillance sentinel sites [8,9]. Briefly, patients suspected to have the above diseases who visited our sentinel sites were included. Starting in March 2020, outpatients presenting with ILI were tested for FLU and SARS-CoV-2, whereas inpatients presenting with SARI were tested for FLU, SARS-CoV-2, and RSV. We then compared the circulation profile, as well as the demographic and clinical characteristics of the FLU and RSV patients before (January 2018 to February 2020) and during the pandemic period (March 2020 to May 2022).

### 2.2. Data and Statistical Analyses

We divided the patients into five age groups: infants and young children (0–4 years), children (5–14 years), adolescents and young adults (15–29 years), adults (30–64 years), and the elderly (≥65 years). For SpS-1, we divided the period according to the three SARS-CoV-2 epidemic waves: wave one (March–September 2020), wave two (March–May 2021), and wave three (December 2021–February 2022). For SpS-2, we considered the pre-pandemic (January 2018–February 2020) and pandemic (March 2020–May 2022) periods. We then compared the proportion of positive cases for each period by taking into account the demographic and clinical variables. Chi-square tests and Fisher exact tests were used to compare the proportion of RV-positive cases during the different periods. We applied the Bonferroni method to adjust the *p-*values if necessary. T-tests and ANOVA were performed to compare the mean ages. *p*-values ≤ 0.05 were considered statistically significant. All the statistical analyses were performed using R (version 4.2.1) software.

## 3. Results

### 3.1. Viral Epidemiology among SARS-CoV-2-Negative Patients during the Epidemic Waves of COVID-19 (SpS-1)

From March 2020 to February 2022, 19% (569/2987) of the tested samples were found to be positive for at least one RV. The proportion of RV-positive samples significantly differed depending on the epidemic wave. The positivity rate was higher in the third wave (25.8%, 258/1001) than in the previous two waves (18.1%, 178/986 and 13.3%, 133/1000 for waves one and two, respectively) (*p* < 0.001) (Table 1).

Among the positive patients, the sex ratio was 1.1 (296 vs. 272) and the mean age was 24.4 years. There was no statistical difference in the mean age of positive patients between the three epidemic waves. Children < 5 years of age (38.6%; 165/428) were more affected by RVs than those aged > 5 years (15.8%, 404/2559) (*p* < 0.001). Conversely, patients aged 30–64 years (13.8%, 193/1401) (*p* < 0.001) and those ≥ 65 (10.1%, 21/207) (*p* = 0.002) were significantly less affected by RV infection during the study period. A comparison of the proportions of positive samples during the epidemic waves showed no substantial difference for the age group < 5 years, unlike the other age groups, for which we noted a higher infection rate during the third wave (Table 1).

Globally, ambulatory patients (22.1%, 401/1812) had a greater RV-positivity rate than those who were hospitalized (14.2%, 126/885) (*p* < 0.001). This difference was noted in every epidemic wave (Table 1).

Concerning the clinical spectrum, 26.3% (242/921) of tested patients with runny noses were positive for RV (*p* < 0.001). In addition, 24.8% (317/1280) and 22.5% (413/1832) of those presenting with fever and cough, respectively, had an RV infection (*p* < 0.001). Vomiting/nausea, sore throat, and headache were reported in 20.2% (26/129), 18.3% (95/520), and 16.3% (128/787) of RV-positive patients, respectively. A comparison of the occurrences of each symptom showed that all but one (myalgia) investigated symptom showed a statistically significant difference between the different waves (Table 1).

Concerning viral etiologies, HRV (6.3%, 187/2987) was the most frequently detected virus overall. Influenza A was identified in 4.1% (121/2987) of tested cases. The remaining pathogens were found at lower proportions (<2%) (Figure 1a, Figure 2). Analysis of the results according to the epidemic wave showed no influenza cases during the first two waves, whereas the virus predominated during the third wave (12.1%, 121/1001) (Figure 1b–d). A similar proportion of HRV cases was observed during the three waves: 6.6% (65/986), 6.8% (68/1000), and 5.4% (54/1001), respectively. CoV-OC43 (3.7%, 36/986) and RSV (3.1%, 31/1000), identified as the second most prevalent viruses during the first and second waves, respectively, also showed the same rate of detection (Figure 1b,c). Of note, the proportion of RSV and CoV-OC43 cases during the first two waves showed overall inverse trends. Viral co-detection occurred in 1.4% (43/2987) of all cases: 1.5% (15/986), 1.4% (14/1000), and 1.4% (14/1001) during the first, second, and third wave, respectively (Appendix A).

### 3.2. Viral Epidemiology among Patients Enrolled through the Influenza Surveillance Network (SpS-2)

During the pandemic period (March 2020 to May 2022), we collected 6297 NP swabs from our ISN, of which 4882 were outpatients and 1415 were inpatients. Overall, the positivity rates for FLU and SARS-CoV-2 were 5.4% (338/6297) and 24.5% (1543/6297), respectively (Table 2). Among the inpatients, RSV was the most detected virus (39.4%, 557/1415). Of note, only the inpatients were screened for RSV (Table 3). Influenza and SARS-CoV-2 were detected in 5.8% (82/1415) and 5.2% (73/1415) of the inpatients, respectively.

The circulation pattern of influenza in Madagascar over the past five years (2018–2022) is shown in Appendix A. Between 2018 and March 2020, influenza circulated throughout the year, in general with periods of higher activity during the rainy season (January–March) and the Austral winter season (June–September) [8,9]. After the first detection of SARS-CoV-2 in Madagascar in March 2020, no influenza virus was detected during the first two waves of SARS-CoV-2. Influenza reappeared in late July 2021, after the second wave of SARS-CoV-2. Indeed, influenza B (Victoria lineage) was detected from July to November 2021, causing a major outbreak, and was then replaced by influenza A (A/H3N2 subtype) between February 2022 and April 2022, causing a second seasonal outbreak. The third wave of SARS-CoV-2 took place between December 2021 and February 2022 (Figure 3).

In the meantime, RSV showed a different pattern of circulation than that of the influenza viruses (Appendix A). Indeed, from 2018 to 2022, RSV showed clear seasonality, with epidemics occurring during the first quarter of the year and reaching a peak between February and March, with a positivity rate of 50–70% for the tested samples (Figure 3 and Appendix A). During the circulation of SARS-CoV-2, RSV continued to circulate in the country. In 2020 and 2021, the RSV epidemic occurred before the first and second SARS-CoV-2 waves, whereas in 2022, the third SARS-CoV-2 wave occurred before the RSV epidemic (Figure 3).

Analysis of the age susceptibility of RSV showed no substantial difference before (mean age = 0.7 years) or during (mean age = 1.2 years) the COVID-19 pandemic (*p* = 0.1). Moreover, there was no statistically significant difference in the proportion of RSV-positive samples by age (*p* = 0.07) or sex (*p* = 0.9) between the two periods (Table 3). Influenza showed a different pattern, although the positive patients identified during the COVID-19 pre-pandemic period were much younger (mean age = 9.1 years) than those identified during the epidemic period (mean age = 14.2 years) (*p* < 0.01) (Table 2). In addition, the proportion of influenza-positive patients by age group was statistically different between the two periods (*p* < 0.001) (Table 3).

Analysis of the age distribution by virus showed three distinct profiles. First, influenza affected all age groups, with a more pronounced infection rate among children (5–14 years), adolescents, and young adults (15–29 years). Second, the RSV infection rate was the highest in children (0–4 years) and declined with age. Finally, the prevalence of SARS-CoV-2 appeared to increase with age, with a higher infection rate observed in adults (30–64 years) (Figure 4).

## 4. Discussion

This study describes the epidemiological profile of RVs before and during the SARS-CoV-2 pandemic in Madagascar. With the exception of RSV, the COVID-19 pandemic influenced the epidemiological trends of all the other investigated respiratory viruses. Indeed, among the 2987 NP specimens negative for SARS-CoV-2 collected from symptomatic patients, only 19% were positive for at least one RV. In earlier studies, we found a higher rate of viral infections (>75%) among patients suffering from either ILI or SARI [8,9]. Although observed at a substantially lower rate than in our previous studies, rhinovirus, the major cause of the common cold, was the most frequently detected virus in this cohort. CoV-OC43 and RSV were the second most frequently detected viruses during the first two epidemic waves of SARS-CoV-2, whereas influenza predominated during the third wave. RSV was not detected in 2020 in this cohort because its inclusion started in March, at the end of the regular circulation period for RSV (first quarter of each year) in Madagascar, as previously described [8,9]. Our results differ from previous observations generated before the COVID-19 pandemic, for which we showed that influenza viruses and RSV were the primary etiologies identified among ILI and SARI patients [8,9]. Nevertheless, they are consistent with those of other studies that reported a significant decline, in particular, of influenza virus, metapneumovirus, parainfluenza virus, and adenovirus during the first waves of COVID-19 [5,6,10,11,12].

Except for HRV, for which the peak of detection appears to coincide with that of SARS-CoV-2, most of the other targeted viruses were detected between the waves of SARS-CoV-2. Madagascar remained “free of influenza” for almost 18 months following the first identification of confirmed SARS-CoV-2 cases. Similar observations were noted in many parts of the globe [13,14,15]. Unlike what has been reported elsewhere, in which a change in the timing of or a delay in RSV epidemics occurred [16,17], RSV continued to propagate during the COVID-19 pandemic throughout the country, with similar amplitude and similar epidemiological characteristics. Of note, the national RSV surveillance is focused primarily on children. Similarly, HRV continued to circulate during the study period, although at a lower rate than what we observed previously, indicating that this virus was not strongly affected by the pandemic. Overall, our study highlighted three different patterns of circulation. The first involved viruses that circulated sporadically but demonstrated greater activity when SARS-CoV-2 was not circulating (human CoV, PIV, HMPV, ADV, and BOV). The second concerned viruses that continued to circulate regardless of the COVID-19 situation (HRV, RSV), and the third involved viruses that were severely disrupted following the spread of SARS-CoV-2 and took longer to reappear (e.g., influenza).

Several factors can explain the observed patterns of RV circulation during the pandemic. The reduction in the circulation of influenza viruses could be the consequence of the early introduction of NPI measures for COVID-19 [6,18,19], including a travel ban. Indeed, a reemergence of influenza viruses occurred at the end of the second SARS-CoV-2 wave (July 2021), in conjunction with the official relaxation of sanitary measures and the resumption of international flights from June 2021. Whether the late 2021 influenza outbreak was due to imported cases is yet to be clarified. Furthermore, a decline in host immunity to new influenza strains cannot be ruled out. Concerning RSV, one of the leading causes of respiratory infections in infants and children [20,21], the similar dynamics of the circulation of this virus during the pandemic could be partially explained by the fact that children under five were not required to wear masks, which allowed RSV to spread within this population. Interestingly, neither the circulation of SARS-CoV-2 nor the implementation of NPI measures influenced the circulation pattern of HRV. A number of authors have postulated that NPI measures may affect enveloped viruses but not non-enveloped viruses [22]. As a non-enveloped virus, HRV is extremely resistant in the environment relative to influenza, making this virus probably more transmissible than enveloped respiratory viruses and probably less affected by NPI measures [23,24,25]. In addition, several studies have suggested that prior HRV infection leads to the inhibition of SARS-CoV-2 replication [26]. Other intrinsic factors could also affect the circulation of other RVs during the pandemic, such as viral interference. Nevertheless, how these mechanisms operate is yet to be determined.

The patients who were positive for RVs were much younger in our current study than the COVID-19-positive patients in our previous study describing the first epidemic wave of SARS-CoV-2 in Madagascar [2] (median age 19.6 vs. 39.3 years, respectively). Host immunity could be a factor in such age susceptibility.

Our study had several limitations. First, during the first months of the pandemic and the different waves, the surveillance capacity was overloaded. As a result, our sentinel sites located throughout the country were unable or found it difficult to ship specimens to the National Influenza Centre, leading to a decrease in the number of specimens collected from suspected ILI and SARI patients at healthcare centers or hospitals. In addition, we observed changes in healthcare-seeking behaviors within communities and at healthcare centers during the first wave of COVID-19, leading to a substantial reduction in the number of medical consultations. Thus, we cannot exclude that we missed a proportion of the sporadic circulation of RVs during these periods, leading to an underestimation of their overall prevalence. Nevertheless, as our study focused on the global circulation of RVs at the community level, we believe that the limitations of our study did not affect the global picture of RV circulation patterns in Madagascar and the observed impact of COVID-19 on them.

## 5. Conclusions

This study confirms the importance of including NPI measures, such as wearing masks, in strategies aimed at reducing the spread of influenza, in addition to vaccination. The impact of NPI measures, in particular, a travel ban during the pandemic, revealed an interesting characteristic of RVs in an insular system such as Madagascar. Indeed, our data suggest that influenza viruses are seeded regularly in the country, leading to seasonal epidemics, but cannot be maintained in the Malagasy population without regular reintroduction. Unlike influenza, RSV and HRV can be maintained within the communities at a local level without requiring annual reintroduction. Therefore, the implementation of RSV vaccination should remain a priority for vulnerable populations.

Our findings underline the importance of an integrated disease surveillance system combined with laboratory surveillance to monitor not only SARS-CoV-2, influenza, and RSV but also other respiratory viruses of interest to public health, with the aim of providing data for decision-making in terms of developing vaccines and therapies [27,28]. Given that SARS-CoV-2 and RVs share a similar clinical spectrum, the integration of COVID-19 into the panel of clinical diagnoses is crucial for better surveillance and monitoring of these viruses.

## Figures and Tables

**Figure 1 viruses-15-00012-f001:**
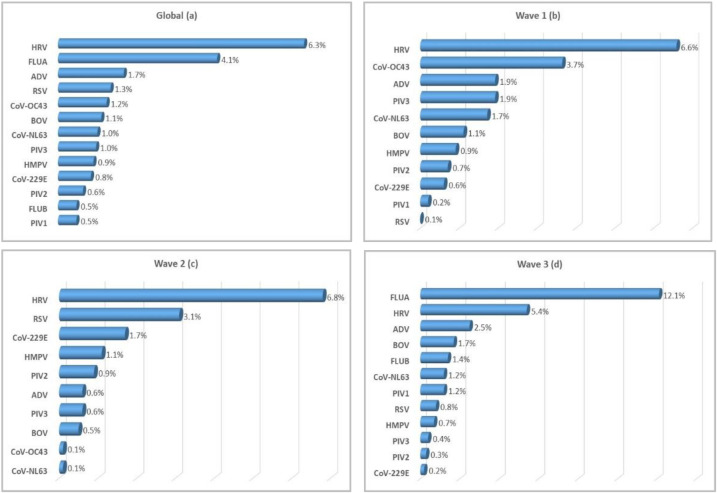
Proportion of viral etiologies detected in COVID-19-negative patients during the three waves (**a**) and during wave 1 (**b**), wave 2 (**c**), and wave 3 (**d**). ADV: adenovirus, BOV: bocavirus, CoV: coronavirus, FLU: influenza, HMPV: human metapneumovirus, HRV: human rhinoviruses, PIV: parainfluenza viruses, RSV: respiratory syncytial viruses.

**Figure 2 viruses-15-00012-f002:**
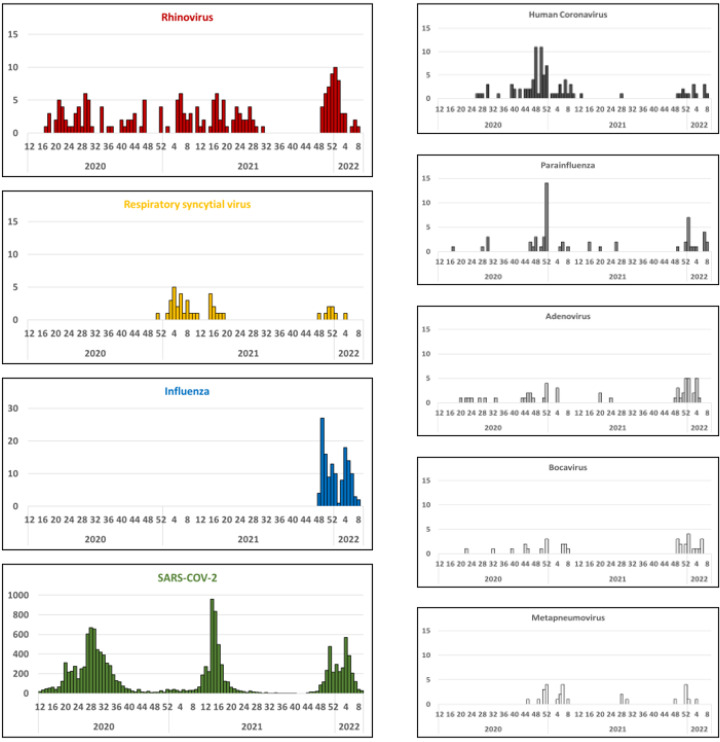
Weekly distribution of respiratory viruses detected in samples from COVID-19-negative patients, March 2020 to February 2022, Madagascar. For comparison, the circulation profile of SARS-CoV-2 during the same period is shown.

**Figure 3 viruses-15-00012-f003:**
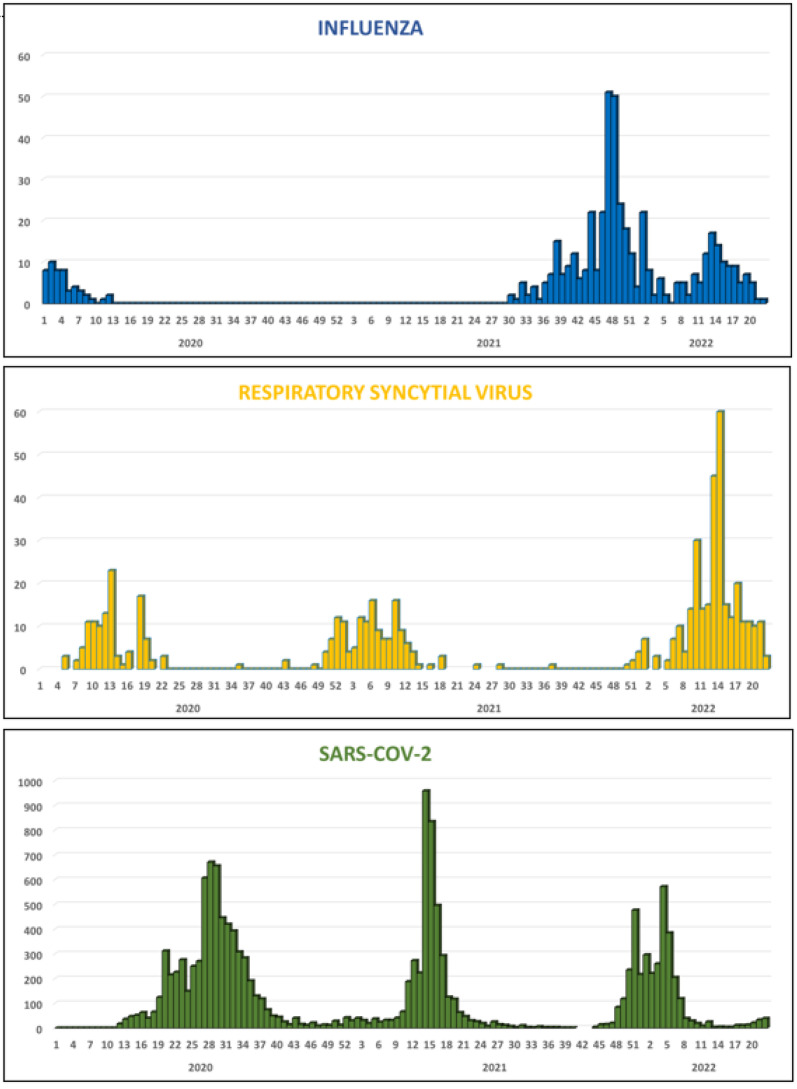
Weekly distribution of influenza, RSV, and SARS-CoV-2 detected by the Influenza Surveillance Network, January 2020 to May 2022, Madagascar.

**Figure 4 viruses-15-00012-f004:**
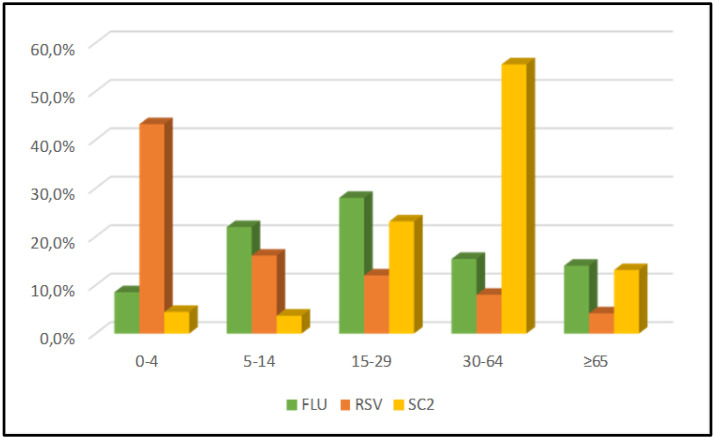
Age distribution of patients who tested positive for influenza (FLU), respiratory syncytial viruses (RSV), or SARS-CoV-2 detected from the Influenza Surveillance Network, January 2020 to May 2022, Madagascar.

**Table 1 viruses-15-00012-t001:** Characteristics of COVID-19-negative patients who tested positive for seasonal respiratory viruses from March 2020 to February 2022.

Variable	Global (Wave 1, 2, and 3)			Wave 1 (Mar-Sep 2020)		Wave 2 (Mar-May 2021)		Wave 3 (Dec 2021-Feb 2022)		
	Tested	Positive	*p*-Value ^3^	Tested	Positive	Tested	Positive	Tested	Positive	*p*-Value *^3^*
	2987	569 (19.0%)		986	178 (18.1%)	1000	133 (13.3%)	1001	258 (25.8%)	<0.001
Mean age (years)	32.8	24.4	<0.001	31.4	24.3	34.3	22.2	32.8	25.7	0.408
Age (years)	N	n (%)		N	n (%)	N	n (%)	N	n (%)	
0–4	428	165 (38.6)	<0.001	139	59 (42.4)	134	44 (32.8)	155	62 (40.0)	
5–14	234	56 (23.9)		79	12 (15.2)	77	18 (23.4)	78	26 (33.3)	0.03
15–29	688	127 (18.5)		262	37 (14.1)	206	24 (11.7)	220	66 (30.0)	<0.001
30–64	1401	193 (13.8)	<0.001	456	61 (13.4)	483	39 (8.1)	462	93 (20.1)	<0.001
≥65	207	21 (10.1)	0.002	48	8 (16.7)	83	3 (3.6)	76	10 (13.2)	0.02
ND	29	7 (24.1)		2	1 (50.0)	17	5 (29.4)	10	1 (10.0)	
Sex										
M	1563	296 (18.9)		500	94 (18.8)	538	68 (12.6)	525	134 (25.5)	<0.001
F	1420	272 (19.2)		484	84 (17.4)	462	65 (14.1)	474	123 (25.9)	
ND	4	1 (25.0)		2	0 (0.0)	0	0 (0.0)	2	1 (50.0)	
Origin										
Ambulatory	1812	401 (22.1)	<0.001	843	159 (18.9)	477	91 (19.1)	492	151 (30.7)	<0.001
Hospital	885	126 (14.2)		134	19 (14.2)	334	25 (7.5)	417	82 (19.7)	
ND	290	42 (14.5)		9	0 (0.0)	189	17 (9.0)	92	25 (27.2)	
Clinical signs ^1^	N			N		N		N		
Cough	1832	413 (22.5)	<0.001	580	123 (21.2)	635	106 (16.7)	617	184 (29.8)	<0.001
Fever	1280	317 (24.8)	<0.001	405	90 (22.2)	432	82 (19.0)	443	145 (32.7)	<0.001
Weakness	1113	161 (14.5)		240	30 (12.5)	362	31 (8.6)	511	100 (19.6)	<0.001
Runny nose	921	242 (26.3)	<0.001	292	58 (19.9)	296	62 (20.9)	333	122 (36.6)	<0.001
Headache	787	128 (16.3)		222	33 (14.9)	268	26 (9.7)	297	69 (23.2)	<0.001
Sore throat	520	95 (18.3)		139	19 (13.7)	174	21 (12.1)	207	55 (26.6)	<0.001
Myalgia	479	70 (14.6)		100	17 (17.0)	166	16 (9.6)	213	37 (17.4)	
Dyspnea	421	56 (13.3)		109	17 (15.6)	173	14 (8.1)	139	25 (18.0)	0.03
Chest pain	289	41 (14.2)		68	8 (11.8	67	4 (6.0)	154	29 (18.8)	0.03
Arthralgia	252	33 (13.1)		130	13 (10.0)	66	5 (7.6)	56	15 (26.8)	0.002
Diarrhea	193	29 (15.0)		30	1 (3.3)	75	5 (6.7)	88	23 (26.1)	<0.001
Vomiting/Nausea	129	26 (20.2)		46	5 (10.9)	20	1 (5.0)	63	20 (31.7)	0.005
Abdominal pain	124	19 (15.3)		25	4 (16.0)	53	3 (5.7)	46	12 (26.1)	0.02
Intercostal recession	68	8 (11.8)		10	1 (10.0)	25	1 (4.0)	33	6 (18.2)	NA
Stridor	51	6 (11.8)		19	1 (5.3)	11	1 (9.1)	21	4 (19.0)	NA
Comorbidities ^2^	185	20 (10.8)	<0.001	112	18 (16.1)	73	2 (2.7)	NA	NA	0.009

ND: not determined, NA: not applicable. ^1^ Number of patients that responded “yes” for a given clinical sign. ^2^ Comparison between waves 1 and 2 only. ^3^ Only significant *p-*values are shown (except for the mean age).

**Table 2 viruses-15-00012-t002:** Characteristics of influenza-positive cases detected among patients enrolled through the Influenza Surveillance Network, Madagascar from January 2018 to May 2022.

	Total		Pre-Pandemic Period		Pandemic Period		*p-*Value ^2^
	Tested	Positive	Tested	Positive	Tested	Positive	
	8792	1076 (12.2%)	2495	738 (29.6%)	6297	338 (5.4%)	<0.001
Mean age (years)	18.9	11.4	7.9	9.1	23.3	14.2	0.01
Age (years)	N	n (%)	N	n (%)	N	n (%)	<0.001
0-4	3532	490 (13.9)	1573	365 (23.2)	1959	125 (6.4)	
5-14	1031	289 (28.0)	457	200 (43.8)	574	89 (15.5)	
15-29	1470	168 (41.9)	227	97 (42.7)	1243	71 (5.7)	
30-64	2242	103 (4.6)	171	59 (34.5)	2071	44 (2.1)	
≥65	179	7 (3.9)	7	3 (42.9)	172	4 (2.3)	
ND	338	19 (5.6)	60	14 (23.3)	278	5 (1.8)	
Sex							
M	4163	535 (12.9)	1267	373 (29.4)	2896	162 (5.6)	<0.001
F	4529	538 (11.9)	1221	363 (29.7)	3308	175 (5.3)	
ND	100	3 (3.0)	7	2 (28.6)	93	1 (1.1)	
Clinical signs ^1^	N		N		N		
Cough	6679	1050 (15.7)	2394	733 (30.6)	4285	317 (7.4)	<0.001
Runny nose	3261	557 (7.1)	1258	398 (31.6)	2003	159 (7.9)	<0.001
Dyspnea	1241	72 (5.8)	273	40 (14.7)	968	32 (3.3)	<0.001
Fever	5788	1015 (17.5)	2354	728 (30.9)	3434	287 (8.4)	<0.001
Intercostal recession	613	16 (2.6)	40	1 (2.5)	573	15 (2.6)	
Weakness	857	50	255	11 (4.3)	602	39 (6.5)	
Myalgia	2601	350 (13.5)	2324	295 (12.7)	277	55 (20.0%)	0.05
Vomiting/Nausea	360	39 (10.8)	95	22 (23.2)	265	17 (6.4)	<0.001
Diarrhea	271	17 (6.3)	65	9 (13.8)	206	8 (3.9)	0.009
Stridor	156	5 (3.2)	28	1 (3.6)	128	4 (3.1)	
Headache	1007	224 (22.2)	379	176 (46.4)	628	48 (7.6)	<0.001
Sore throat	1015	149 (14.7)	329	113 (34.3)	686	36 (5.2)	<0.001
Arthralgia	139	4 (2.9)	1	0 (0.0)	138	4 (2.9)	
Abdominal pain	121	9 (7.4)	11	3 (27.3)	110	6 (5.5)	0.04
Chest pain	136	11 (8.1)	11	5 (45.5)	125	6 (4.8)	<0.001
Asthenia	884	128 (14.5)	219	93 (42.5)	665	35 (5.3)	<0.001
Aggravation	61	0 (0.0)					
Comorbidities	194	9 (4.6)	11	1 (9.1)	183	8 (4.4)	0.4

ND: not determined; ^1^ Number of patients that responded “yes” for a given clinical sign; ^2^ Only significant *p-*values are shown (except for the mean age).

**Table 3 viruses-15-00012-t003:** Characteristics of RSV-positive cases detected among patients enrolled through the Influenza Surveillance Network, Madagascar from January 2018 to May 2022.

	Total		Pre-Pandemic Period		Pandemic Period		*p-*Value ^2^
	Tested	Positive	Tested	Positive	Tested	Positive	
	1816	707 (38.9%)	401	150 (37.4%)	1415	557 (39.4%)	0.5
Mean age (years)	2.9	1.1	1.2	0.7	3.5	1.2	0.1
Age (years)	N	n (%)	N	n (%)	N	n (%)	0.07
0–4	1367	535 (39.1)	357	125 (35.0)	1010	410 (40.6)	
5–14	103	8 (7.8)	15	1 (6.7)	88	7 (8.0)	
15–29	13	3 (23.1)	1	1 (100)	12	2 (16.7)	
30–64	32	1 (3.1)	0	0 (0.0)	32	1 (3.1)	
≥65	11	2 (18.2)	0	0 (0.0)	11	2 (18.2)	
ND	290	158 (54.5)	28	23 (82.1)	262	135 (51.5)	
Sex							
M	925	351 (37.9)	211	79 (37.4)	714	272 (38.1)	0.9
F	799	316 (39.5)	187	71 (38.0)	612	245 (40.0)	
ND	92	40 (43.5)	3	0 (0.0)	89	40 (44.9)	
Clinical signs ^1^	N		N		N		
Cough	1275	459 (36.0)	329	111 (33.7)	946	348 (36.8)	
Runny nose	1073	400 (37.3)	305	107 (35.1)	768	293 (38.2)	
Dyspnea	985	406 (41.2)	191	95 (49.7)	794	311 (39.2)	
Fever	873	304 (34.8)	319	109 (34.2)	554	195 (35.2)	
Intercostal recession	611	233 (38.1)	40	9 (22.5)	571	224 (39.2)	0.05
Weakness	499	158 (31.7)	1	0 (0.0)	498	158 (31.7)	
Myalgia	428	128 (29.9)	323	106 (32.8)	105	22 (21.0)	0.03
Vomiting/Nausea	263	74 (28.1)	47	4 (8.5)	216	70 (32.4)	0.02
Diarrhea	187	49 (26.2)	47	5 (10.6)	140	44 (31.4)	0.009
Stridor	150	46 (30.7)	27	5 (18.5)	123	41 (33.3)	
Headache	116	18 (15.5)	10	1 (10.0)	106	17 (16.0)	
Sore throat	113	28 (24.8)	28	2 (7.1)	85	26 (30.6)	0.02
Arthralgia	97	26 (26.8)	0	0 (0.0)	97	26 (26.8)	
Abdominal pain	87	19 (21.8)	6	1 (16.7)	81	18 (22.2)	
Chest pain	83	10 (12.0)	73	2 (2.7)	10	8 (80.0)	
Asthenia	27	3 (11.1)	24	3 (12.5)	3	0 (0.0)	
Aggravation	61	9 (14.8)	8	2 (25.0%)	53	7 (13.2)	0.03
Comorbidities	369	50 (13.6)	359	50 (13.9)	10	0 (0.0)	NA

ND: not determined, NA: not applicable; ^1^ Number of patients that responded “yes” for a given clinical sign; ^2^ Only significant p-values are shown (except for the mean age).

## Data Availability

Not applicable.

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
