# Peer review of "Epidemiological Patterns of Seasonal Respiratory Viruses during the COVID-19 Pandemic in Madagascar, March 2020–May 2022"

_viruses, 2022, doi:10.3390/v15010012_

Round 1

Reviewer 1 Report

In the current study, the authors identify viral etiologies associated with an acute respiratory infection (ARI) in the context of SARS-CoV-2 circulation during 3 months in Madagascar. Also, they described the epidemiology of respiratory viruses during the pre-pandemic and pandemic period.

Although, the current research may have some local interest otherwise has no new information of interest to the global readership. Though the intention of the authors in this field is good. However, by including some revisions and better interpretations, the work may be accepted as it is relevant to public health.

Comments to authors: 

1) The title is confusing for a reader. Firstly, in the word epidemics, you must describe about "s". Also, the title is not clear to the reader. 

What is your aim? Do you want to describe waves in Madagascar during the COVID-19 epidemic??

2) I understand an article was published previously (DOI: 10.1111/irv.12845). However, could you explain the novelty of the current manuscript? 

3) Epidemic should be added as a keyword in the text. 

4) All keywords should be provided according to MeSH terms at: http://www.nlm.nih.gov/mesh/MBrowser.html

5) All of the abbreviations should be complete at first use in the text and then use the abbreviation. Example: COVID-19, SARS-CoV-2, PCR. 

6) For SARS-CoV-2 the current abbreviation "SC-2" is not suitable. Don't use it.  

7) Line 45: "by real-time RT-PCR 26 415" Not clear. Please check "real-time RT-PCR"?? Alo, 26 415 OR 26415?? 

8) In the introduction and discussion sections I recommend mention to the current test for COVID-19 in Madagascar and also other prevention for COVID-19 done in the current country. 

9) I suggest using this article to improve your introduction section: 

DOI: 10.1016/j.genrep.2021.101417. 

10) Figures 2 and 3 are not clear and not understandable for a reader.  

*) Some of the parts of the manuscript have grammatical errors. 

**) I recommend preparing a graphical abstract to improve the understanding of your study. 

***) All changes in the manuscript should be identified in red font and yellow highlight.

Author Response

REVIEWER 1

In the current study, the authors identify viral etiologies associated with acute respiratory infection (ARI) in the context of SARS-CoV-2 circulation during 3 months in Madagascar. Also, they described the epidemiology of respiratory viruses during the pre-pandemic and pandemic period.

Although, the current research may have some local interest otherwise has no new information of interest to the global readership. Though the intention of the authors in this field is good. However, by including some revisions and better interpretations, the work may be accepted as it is relevant to public health.

Author’s comments: We would like to thank the reviewer for the time spent reviewing our manuscript. We have addressed all comments and consider that our manuscript has improved.

Comments to authors: 

1) The title is confusing for a reader. Firstly, in the word epidemics, you must describe about "s". Also, the title is not clear to the reader. 

What is your aim? Do you want to describe waves in Madagascar during the COVID-19 epidemic??

Author’s comments: The purpose of this study was to describe the impact of SARS-CoV-2/COVID-19 on the circulation of other respiratory viruses over the first two years of the pandemic during which Madagascar had to face three epidemic waves of SARS-CoV-2. As requested by the reviewer, the title was changed for clarity.

2) I understand an article was published previously (DOI: 10.1111/irv.12845). However, could you explain the novelty of the current manuscript? 

Author’s comments: The two papers have different objectives. The previous article (DOI: 10.1111/irv.12845) described the clinical aspects and SARS-CoV-2 laboratory results during the first wave of COVID-19 in Madagascar from March to September 2020. The present manuscript reports the circulation profiles of seasonal respiratory viruses during the entire pandemic period (three first waves of SARS-CoV-2) and compares these circulation patterns between the pre-pandemic and pandemic period. We have modified the abstract and introduction to better explain the objective of the current study.

3) Epidemic should be added as a keyword in the text. 

Author’s comments: As requested, the term "epidemic" has been added.

4) All keywords should be provided according to MeSH terms at: http://www.nlm.nih.gov/mesh/MBrowser.html

Author’s comments: Keywords were adjusted according to MeSH terms

5) All of the abbreviations should be complete at first use in the text and then use the abbreviation. Example: COVID-19, SARS-CoV-2, PCR. 

Author’s comments: Done

6) For SARS-CoV-2 the current abbreviation "SC-2" is not suitable. Don't use it. 

Author’s comments: SARS-CoV-2 was kept all along the manuscript

7) Line 45: "by real-time RT-PCR 26 415" Not clear. Please check "real-time RT-PCR"?? Alo, 26 415 OR 26415?? 

Author’s comments: The sentence has been corrected

8) In the introduction and discussion sections I recommend mention to the current test for COVID-19 in Madagascar and also other prevention for COVID-19 done in the current country. 

Author’s comments: We understand the point of view of the reviewer. Currently, COVID-19 surveillance is integrated into the routine influenza surveillance led by the Institut Pasteur de Madagascar. There’s no particular prevention measure in place so far. We think that there’s no added value to including these pieces of information in particular because measures (NPI and testing) can change at any time pending on the situation. Moreover, this aspect is not connected with the objectives of our study.

9) I suggest using this article to improve your introduction section: 

DOI: 10.1016/j.genrep.2021.101417. 

Author’s comments: We thank the reviewer for his/her suggestion. Nevertheless, in the suggested review, the authors tried to define the immunopathogenesis of SARS-CoV-2 in humans and to provide insight into more effective therapeutic and prophylactic strategies, which is out of the scope of our manuscript. We thus decided not to include it. Nevertheless, considering the comment of the reviewer, we refined the introduction to improve it.

10) Figures 2 and 3 are not clear and not understandable for a reader.  

Author’s comments: Thank you for mentioning that. The two Figures have been improved

*) Some of the parts of the manuscript have grammatical errors.

Author’s comments: The manuscript has been corrected by a native English speaker.

**) I recommend preparing a graphical abstract to improve the understanding of your study. 

Author’s comments: A graphical has been added in Supplementary Figure S1

***) All changes in the manuscript should be identified in red font and yellow highlight.

Author’s comments: The mode track-change is used to follow and identify revisions

Reviewer 2 Report

In this study, Razanajatovo et al. have investigated the circulation of respiratory viruses between March 2020 and May 2022 in the context of SARS-CoV-2 pandemics. The samples were on the one hand, samples from patients who tested negative for SARS-CoV-2. Secondly, samples were collected at the Influenza sentinel network in order to compare circulation of different respiratory viruses including Sars-CoV-2.

Overall, the study is well designed and presented and the results are very interesting for readers.

I have not found anything to be changed.

Author Response

REVIEWER 2

Comments and Suggestions for Authors

In this study, Razanajatovo et al. have investigated the circulation of respiratory viruses between March 2020 and May 2022 in the context of SARS-CoV-2 pandemics. The samples were on the one hand, samples from patients who tested negative for SARS-CoV-2. Secondly, samples were collected at the Influenza sentinel network in order to compare circulation of different respiratory viruses including Sars-CoV-2.

Overall, the study is well designed and presented and the results are very interesting for readers.

I have not found anything to be changed.

Author’s comments: We thank the reviewer for his encouraging comments. We have made some changes in the revised version that have greatly improved our manuscript.

Round 2

Reviewer 1 Report

accept